# Community-Centered Assessment to Inform Pandemic Response in Georgia (US)

**DOI:** 10.3390/ijerph20095741

**Published:** 2023-05-08

**Authors:** Tabia Henry Akintobi, Rakale C. Quarells, Robert A. Bednarczyk, Saadia Khizer, Brittany D. Taylor, Michelle N. A. Nwagwu, Mekhi Hill, Claudia E. Ordóñez, Gaëlle Sabben, Sedessie Spivey, Kayla Davis, Michael L. Best, Amy Z. Chen, Katherine Lovell, Leslie S. Craig, Mohamed Mubasher

**Affiliations:** 1Community Health and Preventive Medicine, Morehouse School of Medicine, Atlanta, GA 30301, USA; 2Rollins School of Public Health, Emory University, Atlanta, GA 30322, USA; 3Vaccination Trial Unit, Community Health and Preventive Medicine, Morehouse School of Medicine, Atlanta, GA 30301, USA; 4DeKalb County Board of Health, Decatur, GA 30031, USA; 5Sam Nunn School of International Affairs and the School of Interactive Computing, Georgia Institute of Technology, Atlanta, GA 30308, USA; 6Southside Medical Center, Atlanta, GA 30315, USA; 7Independent Researcher, Bridgetown 15001, Barbados

**Keywords:** community-powered research, quantitative data, public health response, disproportionately impacted communities

## Abstract

The Georgia Community Engagement Alliance (CEAL) Against COVID-19 Disparities Project conducts community-engaged research and outreach to address misinformation and mistrust, to promote inclusion of diverse racial and ethnic populations in clinical trials and increase testing and vaccination uptake. Guided by its Community Coalition Board, The GEORGIA CEAL Survey was administered among Black and Latinx Georgia 18 years and older to learn about community knowledge, perceptions, understandings, and behaviors regarding COVID-19 testing and vaccines. Survey dissemination occurred using survey links generated through Qualtrics and disseminated among board members and other statewide networks. Characteristics of focus counties were (a) highest proportion of 18 years and older Black and Latinx residents; (b) lowest COVID-19 testing rates; and (c) highest SVI values. The final sample included 2082 surveyed respondents. The majority of participants were men (57.1%) and Latinx (62.8%). Approximately half of the sample was aged 18–30 (49.2%); the mean age of the sample was 33.2 years (SD = 9.0), ranging from 18 to 82 years of age. Trusted sources of COVID-19 information that significantly predicted the likelihood of vaccination included their doctor/health care provider (*p*-value: 0.0054), a clinic (*p*-value: 0.006), and university hospitals (*p*-value: 0.0024). Latinx/non-Latinx, Blacks vs. Latinx, Whites were significantly less likely to get tested and/or vaccinated. Non-Latinx, Blacks had higher mean knowledge scores than Latinx, Whites (12.1 vs. 10.9, *p* < 0.001) and Latinx, Blacks (12.1 vs. 9.6, respectively, *p* < 0.001). The mean knowledge score was significantly lower in men compared to women (10.3 vs. 11.0, *p* = 0.001), in those who had been previously tested for COVID-19 compared to those who had never been tested (10.5 vs. 11.5, respectively, *p* = 0.005), and in those who did not receive any dose of vaccination compared to those who were fully vaccinated (10.0 vs. 11.0, respectively, *p* < 0.001). These data provide a benchmark for future comparisons of the trajectory of public attitudes and practices related to the COVID-19 pandemic. They also point to the importance of tailoring communication strategies to specific cultural, racial, and ethnic groups to ensure that community-specific barriers to and determinants of health-seeking behaviors are appropriately addressed.

## 1. Introduction

The SARS-CoV-2 or Coronavirus Disease 2019 (COVID-19) pandemic is the largest public health crisis in over a century. COVID-19 has exposed many vulnerabilities in our health and educational systems, the environment, and the global economy at large. As the severity of the pandemic increased in 2020 and 2021, mitigating the risk of transmission and infection through improving the public’s knowledge of COVID-19 and COVID-19 vaccines became paramount. COVID-19 infection rates, general trust and/or mistrust in vaccines, and the health system, and COVID-19 vaccine hesitancy all raised concerns about the potential challenges of vaccine implementation and resulting hurdles to gaining control over the COVID-19 pandemic. These concerns are of the utmost importance for Black/African American and Latinx/Hispanic (also referred to as Latino and Latina) (hereafter called Black and Latinx) populations, where COVID-19-related health disparities and vaccine hesitancy are highest [1].

Racial/ethnic minority populations have historically borne a disproportionate burden of illness, hospitalization, and death during public health emergencies, including the 2009 H1N1 influenza pandemic and the Zika virus epidemic [2,3,4,5]. Research has shown that overall, Black, Latinx, American Indian and Alaska Native, and Asian, Native Hawaiian, and Other Pacific Islander people fare worse compared to White people across most examined measures of social determinants of health for which data were available [6]. These characteristics include, but are not limited to, discrimination, low socioeconomic status and power, predisposing racial/ethnic minority populations in general and Blacks in particular to less-than-optimal living conditions and quality of life. Racial/ethnic minority populations are more likely than non-Latinx White populations to live in densely populated areas, overcrowded housing, and/or multigenerational homes; lack adequate plumbing and access to clean water; and/or have jobs that do not offer paid leave or the opportunity to work from home [7,8]. In the context of COVID-19, these factors have contributed to a person’s ability to comply with pandemic mitigation mandates established to reduce risk of infection, such as physical distancing and sheltering in place [9].

While public health crises-affiliated disparity statistics explain differences in illness, hospitalization and death, they do not advise or inform communication, education and outreach strategies that can be used for response strategies towards recovery. The significance of leveraging the leadership, trust, and networks of community-based organizations to address persistent health issues and emerging health crises are among critical approaches that have been considered to develop and deploy action strategies. However, these strategies are nuanced and may differ by priority populations as well as by each crisis itself.

The Garnering Effective Outreach and Research in Georgia for Impact Alliance (GEORGIA)—Community Engagement Alliance (CEAL) Against COVID-19 Disparities Project (hereafter called GEORGIA CEAL) was funded as part of a 21-state network to conduct urgent community-engaged research and outreach focused on COVID-19 awareness and education to address misinformation and mistrust and promote and facilitate inclusion of diverse racial and ethnic populations in clinical trials along with vaccination uptake. Formed in September 2020 and funded by the National Institutes of Health (NIH), this nationwide initiative is an ongoing effort to provide trustworthy information through active community engagement and outreach to the people hardest hit by the COVID-19 pandemic, including Blacks, Latinx, and American Indians/Alaska Natives, with the goal of building long-lasting partnerships as well as improving diversity and inclusion in the research response to COVID-19 [10,11].

The GEORGIA CEAL Community Coalition Board (CCB) governs all project efforts and is composed of a state-representative, community-majority to ensure that research and outreach processes and findings are translated with, co-created by, and relevant to communities towards effective COVID-19 education, public health communication, research, and sustainability of evidence-based approaches. The GEORGIA CEAL Survey was among the community-driven research strategies implemented to inform action strategies. Its purpose was to learn about community knowledge, perceptions, understandings, and behaviors regarding COVID-19, vaccines against COVID-19, and vaccine research trials; compare and contrast findings for priority population groups and use findings to inform and develop COVID-19 response strategies (i.e., outreach, education, health promotion, communication) towards vaccine trial participation and vaccination uptake and population health equity strategies, at large. The purpose of this article is to detail processes, initial findings, and implications associated with the GEORGIA CEAL survey.

## 2. Materials and Methods

### 2.1. Survey Instrument Development

Both the national CEAL Assessment and Evaluation Workgroup (A and EW) and the GEORGIA CEAL’s CCB were instrumental in the creation of the GEORGIA CEAL Common Survey 1. Common Survey items were established to support a systematic approach to data collection and sharing across the 21-state CEAL Alliance which set the stage for a multi-layered evaluation. Additionally, CEAL Survey translations were made available for use by CEAL Alliance members. CEAL Alliance members also had the freedom to include additional items based on the input of their community partners. Data sharing among the CEAL Alliance and community-level analyses were seen as critical to understanding the full impact of the CEAL efforts.

Initially, the national CEAL A and EW collaborated to create the CEAL Alliance-wide survey that assessed demographics, social determinants of health, COVID-19 prevention behaviors, participation in COVID-19 vaccine research trials, and perceptions of COVID-19 vaccine confidence and uptake. The GEORGIA CEAL research team and CCB members worked together to review the CEAL Alliance A and EW Surveys (English and Spanish) to determine appropriateness for the communities served in Georgia. Suggested changes were made to English and Spanish item wording for better understanding by community members. The GEORGIA CEAL research team and CCB members decided on additional survey items of interest, including trusted messengers, misinformation, and additional items related to social determinants of health.

Once the GEORGIA CEAL Surveys were finalized and approved by the CCB, the survey administration process was approved by the MSM institutional review board (Study #1664429-25). Survey dissemination occurred using survey links generated through Qualtrics [12]. These links were distributed through GEORGIA CEAL CCB members and Morehouse School of Medicine (MSM) community partner networks not directly associated with GEORGIA CEAL. Community partners shared the survey links throughout their networks, i.e., through newsletters, email listservs, organizations’ social media accounts, and websites. All survey data collection occurred online. Surveys were administered April–June 2021.

### 2.2. Sampling Plan

The underlying sampling method was based on a two-stage stratified-cluster sampling proportionate-to-county-size/weights approach, with county as the cluster unit for sampling. The size/weights were derived from county-level total counts of Black and Latinx residents aged 18 years and older, vaccination rates, and the CDC-developed Social Vulnerability Index (SVI) [6]. The intent was to obtain a population-based representative sample of Black and Latinx residents disproportionately impacted by adverse health and health care outcomes, in general, and specifically, by COVID-19. Characteristics of focus counties were (a) highest proportion of 18 years and older Black and Latinx residents; (b) lowest COVID-19 testing rates; and (c) highest SVI values. These factors were used to stratify counties and inform sampling. Threshold values were set based on county-specific statistical distributions for the three stratification factors. To mathematically formulate this approach, we borrowed two linear algebra concepts of a “*vector*” (collection of values) and the “*norm*” (length/magnitude of a vector). For the first phase of the two-stage cluster sampling plan, we formed a 4-elements vector of data for each Georgia county (n = 159). Each vector was composed of (a) the proportion of Black residents aged 18 years and older; (b) the proportion of Latinx residents aged 18 years and older; (c) the reciprocal of vaccination rates; and (d) the county SVI. In order to determine which counties (represented by these vectors of the four components) ranked highest, i.e., had the highest proportions Black/Latinx adult residents and SVI values as well as lowest vaccination rates, we calculated the norm values of these vectors (the norm of a vector is simply the square root of the sum of each component squared). We consequently ranked Georgia counties in an ascending order and randomly selected 19 counties from the top 25 counties.

#### Sample Size and Precision of Estimates Considerations

The survey sampling plan was designed to estimate:Characteristics of the survey respondentsCOVID-19 testing and/or vaccination behaviorsCOVID-19 knowledgeAttitudes and behavior towards testing and vaccinationIntent to vaccinateLevel of trust towards personal, health/medical, state, and federal entities

We selected level of knowledge (% of correct answers) as an outcome of interest to drive estimates of sample size that would secure a minimal margin of error from a targeted level of knowledge/trust as well as vaccination/testing rates, using the 95% confidence interval approach. As indicated above, for the first phase of cluster sampling we appealed to the concept of the “norm” and vectors to rank the 159 Georgia counties. For the second stage of our cluster sampling process, we first summed all the norm values for the selected counties. Next, for each county, we calculated the magnitude of their “norm” value relative to the sum of norm values for all selected counties. That relative magnitude value was used as a weight to distribute our required total sample size of 2004 responders across counties. We thus also determined the county-specific counts of needed respondents based on a proportional-to-size approach. The total sample size of 2004 ensured a maximum margin of error of 5–7% for the expected level of knowledge, proportion trusting the health system (completely/ versus no trust), and the proportion who got tested and those vaccinated against COVID-19, respectively. The total sample size goal of 2004 ensured a maximum margin of error of 5–7% for the expected level of knowledge, proportion trusting the health system (great deal of trust versus no trust), and the proportion who got tested and those vaccinated against COVID-19, respectively. The final sample included 2082 surveyed respondents.

### 2.3. Statistical Analyses

Characteristics of survey respondents were summarized by the mean and standard deviations (SDs) for continuous variables and frequency and percentages for categorical variables. A summary knowledge score, based on responses to each of 18 individual questions, was created by summing correct/incorrect responses so that higher scores reflected greater knowledge (i.e., correct responses were coded as ‘1’ and incorrect responses (including “don’t know”) were coded as ‘0’). Analyses for specific outcomes (e.g., summary knowledge) were restricted to respondents with complete data for that analysis; therefore, effective sample sizes differ by analysis, and are presented for each specific analytic component. Generalized Linear Mixed Models were applied using the logit link with county of residence as a random effect, controlling for socio-demographic factors (i.e., age, race, ethnicity, gender, education level, and employment status), separately examined factors associated with being tested for COVID-19 and being fully vaccinated against COVID-19. All analyses were performed using SAS version 9.4 and Stata v15 [13]. The overall significance level was set at 0.05 and Holm–Bonferroni method was used whenever needed to control for multiple comparisons and inflation of type I error [14]. Imputation of missing data was applied to survey knowledge and trust questions based on identifying (non-missing) predictors of missing data using procedures Multiple Imputations and Survey Impute in SAS [15].

## 3. Results

### 3.1. Sample Characteristics

Only adult men and women who self-identified as Black (Latinx and non-Latinx) or Latinx (White, Black) were included in the analytic sample (n = 2037). Analyses were further restricted to those who were not missing information on key demographics (e.g., age, education, employment status) and outcomes of interest (i.e., prior experience of COVID-19 testing and vaccination). There were no statistically significant differences between those with complete (n = 2001) versus missing (n = 36) data with respect to education level and employment). However, those with complete data were more likely to be younger (*p* = 0.01), men (*p* < 0.001), self-identify as Latinx and White (Latinx, White; n = 729; *p* < 0.05), to have previously tested for COVID-19 (*p* < 0.05) and be fully vaccinated (i.e., completed vaccination primary series) (*p* < 0.05). The final full analytic sample of 2001 respondents included those who self-identified as non-Latinx and Black (non-Latinx, Black; n = 744), Latinx and Black (Latinx, Black; n = 729), and Latinx, White (n = 528). Analysis for trusted sources of information was conducted on a sample of 1441 individuals with complete data (560 Latinx, Black; 454 Latinx, White; 427 non-Latinx, Black).

Per Table 1, the majority of participants were men (57.1%), of Black/AA race (73.6%) and of Latinx ethnicity (62.8%). Approximately half of the sample was aged 18–30 (49.2%); the mean age of the sample was 33.2 years (SD = 9.0), ranging from 18 to 82 years of age. The majority of participants (74.2%) had a college-level education or higher and were employed (i.e., 87.9% either working for pay [full-time or part-time] or working without pay [e.g., as an intern]). Further, the majority of respondents (87.2%) had been previously tested for COVID-19, with 6.2% reporting having tested positive. The sample included 47.5% who were fully vaccinated, 20.2% partially vaccinated and 32.3% who had received no dose of vaccination. Sample characteristics, by race and ethnicity, are described in Table 1.

#### Participants’ Characteristics by Ethnicity/Race

Fisher/Exact testing indicated statistically significant disparity (*p*-value < 0.001) in the frequency distribution of ethnicity/race (Hispanic White/Non-Hispanic Black/Hispanic Black) by each of the participants’ characteristics (Table 1)

### 3.2. COVID-19-Related Trusted Source of Information and Knowledge

#### 3.2.1. Trusted Sources of Information

Among the 1441 with non-missing data on trusted sources of information more than half reported a great deal of trust in the US Coronavirus Task Force (61.1%), doctors or health care providers (54.0%), and close friends/family members (51.7%). However, top trusted sources of information varied across racial and ethnic groups with the US Coronavirus Task Force being most often cited as trusted “a great deal” among Latinx, Whites (60.4%) and Latinx, Blacks (67.3%), while doctors or health care providers were most highly regarded among non-Latinx, Blacks (56.4%). Trusted sources of information, by race and ethnicity, are described in further detail in Table 2.

#### 3.2.2. Trusted Sources by Ethnicity/Race

Fisher/Exact testing indicated statistically significant disparity (*p*-value < 0.037) in the frequency distribution of ethnicity/race (Hispanic White, Non-Hispanic Black, Hispanic Black) by each of the trust sources (Table 2)

#### 3.2.3. Predictors of COVID-19 Trusted Sources of Information

Some of the findings displayed on Table 3 identifies that participants who are (1) younger and/or of women gender are most trusting of their doctors/health care providers (*p*-value < 0.0078), (2) of high school/less education are most trusting of their faith leaders (*p*-value = 0.0008), (3) younger are most trusting of their friends (*p*-value = 0.0038) but Blacks/AA and women are least trusting of their friends (*p*-value < 0.0001), and (3) Blacks/AA and most educated are least trusting to the news/media. The rest of the findings are displayed on Table 3.

### 3.3. COVID-19 Related Knowledge

Of the full sample, 893 (44.6%) had non-missing responses for all 18 knowledge questions about COVID-19 (400 Latinx Black; 295 Latinx, White; 198 non-Latinx Black). There were no statistically significant differences between those with complete (n = 893) versus missing data (n = 1144) with respect to age, sex, employment, and vaccination status (*p* < 0.05). However, those with complete data were more likely to have a college or higher education (*p* = 0.008), self-identify as Latinx, White (*p* < 0.001) than non-Latinx, Black, and to have previously tested for COVID-19 (*p* < 0.001).

The overall mean knowledge score was 10.6 of 18 (SD=3.2, range between 2–18), with variation by race/ethnicity. Non-Latinx, Blacks had higher mean scores than Latinx, Whites (12.1 vs. 10.9, *p* < 0.001) and Latinx, Blacks (12.1 vs. 9.6, respectively, *p* < 0.001). The mean knowledge score was significantly lower in men compared to women (10.3 vs. 11.0, *p* = 0.001), in those who had been previously tested for COVID-19 compared to those who had never been tested (10.5 vs. 11.5, respectively, *p* = 0.005), and in those who did not receive any dose of vaccination compared to those who were fully vaccinated (10.0 vs. 11.0, respectively, *p* < 0.001). There was no statistically significant difference in mean knowledge scores between those who reported partial versus full vaccination (*p* = 0.089) or those who were partially vaccinated compared to those who had received no dose of vaccination (*p* = 0.077).

#### Predictors of COVID-19 Related Knowledge

Table 4 displays the correct responses to knowledge questions that have statistically significant predictors. It shows that younger participants were more likely to recognize the potential protection of the vaccine (*p*-value < 0.0002) and misinformation about possible cures (*p*-value ≤ 0.0001). The table also indicates that those who correctly answered the question about the possible sickness of the vaccine itself were the Latinx/non-Latinx, Blacks (*p*-value = 0.003) and those who are highly educated (*p*-value <0.0272).

### 3.4. COVID-19 Testing and Vaccination by Key Covariates

#### Knowledge and Trust Vis-a-Vis Testing and Vaccination

Relative to those who were tested/vaccinated, those who were not, displayed a significantly lower level of knowledge in terms of risks of COVID-19, vaccine safety/efficacy and COVID-19 mitigating hygienic practices/mode of transmission (Table 5). COVID-19 infection and death rates were higher among Latinx/non-Latinx, Blacks (*p*-value: 0.0089). Trusted sources of COVID-19 information that significantly predicted the likelihood of vaccination included their doctor/health care provider (*p*-value: 0.0054), a clinic (*p*-value: 0.006), university hospitals (*p*-value: 0.0024). Latinx/non-Latinx, Blacks vs. Latinx, Whites were significantly less likely to get tested and/or vaccinated. Other predictors of testing and/or vaccination included those who have a higher degree of perception of COVID-19 risk, those who had health insurance, those who had a college education or higher, and those who were employed vs. unemployed.

## 4. Discussion

Utilizing a survey of over 2000 Black and Latinx adults in Georgia counties with high proportions of Black and/or Latinx populations, and low COVID-19 testing and vaccine uptake, we have characterized COVID-19 mitigation knowledge, attitudes, and practices during the second year of the pandemic and amidst early broad-scale uptake of emergency use authorized vaccinations. These data provide a benchmark for future comparisons of the trajectory of public attitudes and practices related to the COVID-19 pandemic. Identification of trusted sources of information and factors associated with COVID-19 testing and vaccination were the two key outcomes in this analysis.

In GEORGIA CEAL, survey participants reported having the most trust in medical professionals, including their own health care providers and the U.S. Coronavirus Task Force, as well as their close friends and family members. These findings echo previous research including a survey of Black and Latinx communities in Pittsburgh which documented that community members trusted medical and scientific personnel the most, especially local doctors and the county health department [16,17,18]. Implications reflect the significant need for health care providers to act as spokespersons for health-promoting and disease-preventing measures. They should be equipped to effectively communicate and to leverage culturally tailored communication strategies both within and outside of the clinical setting.

We also found that participants had the least trust in their social media contacts and faith leaders for COVID-19 information, which also aligns with the findings of Ragavan et al. [19], which found that while social media trust was low, there was high utilization of social media channels, with 72% of their participants reporting use of Facebook as a source of information-higher than any other source asked about in the study. In our study, 34% of all participants reported putting a great deal of trust in faith leaders. This finding contrasts with a previous study that found only 13% of participants report using faith leaders as trusted sources of information, though 89% of that subset had a high degree of trust in faith leaders. Given that Georgia is in the “Bible Belt” and historically Black Americans have reported being more religious than other groups [20], the relatively small proportion of individuals reporting a great deal of trust in their faith leaders in our study may be due to the lower mean age of participants in the survey (33.2 years). This would align with other studies which have reported that religiosity and related trust increases with age [21].

### 4.1. COVID-19-Related Knowledge

Our findings also indicated statistically significant differences in COVID-19 knowledge by race/ethnicity and gender. First, both Latinx, Black and Latinx, White identifying respondents had lower mean knowledge scores when compared to non-Latinx, Blacks. This could be attributed to the slower national pace in development and dissemination of translated and culturally tailored COVID-19 communication designed for Latinx communities. Further, Latinx, White participants had higher knowledge scores compared to Latinx, Blacks potentially reflecting the distinctly more positive health/health care lived experience of White versus Black participants, despite having ethnicity in common [22]. Men having lower mean knowledge scores than women align with other studies and reviews indicating stronger health information-seeking behaviors among women compared to men [23].

### 4.2. Vaccination Predictors

With respect to predictors of vaccination, vaccine hesitancy and confidence, we found, among the population we surveyed, Blacks, irrespective of ethnicity, were less likely to have been vaccinated compared to others. While our survey sample was not designed to be representative of the general population of Georgia, documenting low COVID-19 vaccine uptake in this population supports prior findings of racial disparities in COVID-19 vaccination, and may result from historical maltreatment and medical racism. Previous studies of COVID-19 vaccination hesitancy in Latinx and Blacks found that the major predictors of vaccine hesitancy included sociodemographic characteristics (e.g., younger age, women, lower-income/education, and larger household size); medical mistrust and history of racial discrimination; greater exposure to myths and misinformation; perceived risk of getting infected with COVID-19; past vaccine compliance and beliefs about vaccines; and concerns about the safety, efficacy, and side effects from the COVID-19 vaccines [24]. Moreover, in nationally representative surveys, both Woko et al. [25] and Kreps et al. [26] found Black Americans to have a lower intent to receive the COVID-19 vaccination due to more negative perceptions about the vaccination (such as believing that it would cause severe side effects) compared to other race/ethnic groups. These surveys, however, were both conducted before wide availability of the COVID-19 vaccines, and thus only assessed vaccination intention, while our analysis included reported vaccinations. A recently published scoping review found that age, gender, education level, race/ethnicity, among other factors, were significantly associated with intentions to accept the COVID-19 vaccine [27]. In an investigation of factors associated with the intention to obtain the COVID-19 vaccination, conducted during the same timeframe as our study, Blacks and Latinx survey respondents were also significantly less likely to report that they would get vaccinated [28]. Analyses controlling for beliefs about vaccine safety and efficacy resulted in insignificant racial and ethnic differences. These results, along with those of our study, signal that community outreach efforts to address vaccine misinformation, in the context of historical injustices and efforts to address these, will be critical to improve protection against COVID-19 in racial and ethnic minorities in Georgia by reducing vaccine hesitancy and increasing confidence and vaccine uptake.

In considering COVID-19 knowledge (including beliefs/misconceptions/misinformation) as predictors of vaccination, we found that accurate knowledge related to COVID-19 transmission and effectiveness of protective measures was most predictive of COVID-19 vaccination. These findings underscore the need for effective and culturally appropriate community-based education programs related to COVID-19 mitigation, utilizing trusted sources of information.

### 4.3. Strengths and Limitations

This study has noteworthy strengths and limitations. Notably, the dissemination of the survey through the GEORGIA CEAL’s CCB provided a means to work with established community partners who are trusted by their constituencies, offering a way to achieve rapid survey dissemination and completion by community members. The findings from this survey are not generalizable to the full population of Georgia, as we focused on counties with high proportions of Black and Latinx populations that were deemed at highest risk of poor COVID-19-related outcomes. Therefore, the findings in this report may reflect individuals with different experiences during the pandemic compared to the general population. However, as these minority populations have experienced a disproportionate burden of COVID-19, it is appropriate to focus specifically on these groups.

These data were collected April–June 2021 and the information, public perception and research environment have rapidly evolved over the course of the COVID-19 pandemic, as has the widespread availability of vaccines against COVID-19, including updated boosters. It is worth noting that the survey tools used for capturing data for predictors of COVID-19 vaccination/testing in this investigation were not intended to measure psychological variables/constructs that are necessary for delineating attitudes towards vaccination/testing hesitancy or willingness, as detailed elsewhere [29]. Recent and novel advances in analytic approaches towards precision in assessing the role of implicit vaccination attitudes and their effects on hesitancy and corresponding behaviors were not applied in our study and should be considered in subsequent investigations [30]. Other investigations of the role of conspiracy theories and their psychological effects on the likelihood to get the COVID-19 vaccination, recently conducted in other countries, should be adapted in the US, towards assessment of consistency in outcomes, potentially informing proactive and more strategic individual- and community-level public health intervention and communication strategies [31]. While these findings may not reflect the current state of COVID-19 mitigation among Black and Latinx populations in Georgia, it does provide a baseline from which future studies can evaluate their findings to identify shifting public perceptions, particularly among younger Black and Latinx residents, with nuanced and unique experiences of racism, health care discrimination and challenges navigating health and health care systems. Results contribute to a growing roadmap comprehensively reflecting the science and art of preparing for and proactively acting to intervene among those disproportionately experiencing the negative consequences of public health pandemics. Individual and contextual factors influencing their vaccination decisions and the communication strategies developed to promote prevention must be understood towards optimal, community-centered response.

## 5. Conclusions

These results point to the importance of tailoring communication strategies to specific cultural, racial, and ethnic groups to ensure that community-specific barriers to and determinants of health-seeking behaviors are appropriately addressed. This is particularly important in the context of public health emergencies when the adoption of prevention behaviors and practices by disproportionately at-risk communities is crucial to minimize the impact of the emergency on those communities and the population at large. Beyond developing interventions and communication strategies to respond to emerging public health needs, it is imperative that we continue to engage community organizations and members of historically marginalized communities in efforts to track and understand responses to public health messages, as well as strengthen trust in reliable messengers.

The GEORGIA CEAL program is well-positioned to use these findings to improve the health of Black and Latinx Georgians, by promoting COVID-19 vaccination and assisting in addressing mis- and dis-information about COVID-19. This network of partners has a wide reach regarding health-related topics and can continue to serve minority populations in Georgia to address gaps in health equity, for COVID-19 as well as other conditions where health disparities exist.

## Figures and Tables

**Table 1 ijerph-20-05741-t001:** Sample Characteristics, by Race and Ethnicity.

Characteristics	Hispanic, White (n = 528)	Non-Hispanic, Black (n = 744)	Hispanic, Black (n = 729)	Total (n = 2001)	*p*-Value
	n (%)	n (%)	n (%)	n (%)	
**Age-group**					
18–30 years	292 (55.3)	279 (37.5)	414 (56.8)	985 (49.2)	**<0.001**
31–40 years	199 (37.7)	293 (39.4)	260 (35.7)	752 (37.6)	
Older than 40 years	37 (7.0)	172 (23.1)	55 (7.5)	264 (13.2)	
**Sex**					
Man	330 (62.5)	320 (43.0)	493 (67.6)	1143 (57.1)	**<0.001**
Woman	198 (37.5)	424 (57.0)	236 (32.4)	858 (42.9)	
**Education level**					
High school or less	158 (29.9)	162 (21.8)	197 (27.0)	517 (25.8)	**0.003**
College or higher	370 (70.1)	582 (78.2)	532 (73.0)	1484 (74.2)	
**Employment** **status**					
Unemployed ^¥^	58 (11.0)	112 (15.1)	72 (9.9)	242 (12.1)	**0.006**
Employed	470 (89.0)	632 (85.0)	657 (90.1)	1759 (87.9)	
**Ever tested for COVID-19**					
No	49 (9.3)	135 (18.2)	72 (9.9)	256 (12.8)	**<0.001**
Yes	479 (90.7)	609 (81.9)	657 (90.1)	1745 (87.2)	
**Vaccination status**					
No dose of vaccination	100 (18.9)	234 (31.5)	312 (42.8)	646 (32.3)	**<0.001**
Partially vaccinated	105 (19.9)	131 (17.6)	169 (23.2)	405 (20.2)	
Fully vaccinated	323 (61.2)	379 (50.9)	248 (34.0)	950 (47.5)	
**Total household income** ** ^ǂ^ **					
Less than USD 35,000	153 (29.1)	206 (27.7)	192 (26.4)	551 (27.6)	**<0.001**
USD 35,000–<USD 50,000	205 (39.1)	179 (24.1)	271 (37.2)	655 (32.8)	
USD 50,000–<USD 75,000	141 (26.9)	224 (30.1)	169 (23.2)	534 (26.7)	
≥USD 75,000	24 (4.6)	114 (15.3)	91 (12.5)	229 (11.5)	
Prefer not to answer	2 (0.4)	21 (2.8)	5 (0.7)	28 (1.4)	
**Perceived risk** **of getting** **COVID-19 ^ǂ^**					
High risk	37 (7.1)	66 (8.9)	56 (7.7)	159 (8.0)	**<0.001**
Moderate risk	162 (30.9)	238 (32.0)	177 (24.3)	577 (28.9)	
Low risk	218 (41.5)	304 (40.9)	290 (39.8)	812 (40.6)	
No risk	84 (16.0)	58 (7.8)	91 (12.5)	233 (11.7)	
Do not know /No opinion	14 (2.7)	47 (6.3)	103 (14.1)	164 (8.2)	
Not applicable /Refused	0 (0.0)	14 (1.9)	7 (1.0)	21 (1.1)	
I’ve previously tested positive for COVID-19	10 (1.9)	17 (2.3)	5 (0.7)	32 (1.6)	

^¥^ Unemployed includes those who self-reported as being unemployed, on leave, retired, staying at home, student and/or unable to work because of a disability. **^ǂ^** <2% missing data on household income and perceived risk of getting COVID-19. Bold indicates statistical significance.

**Table 2 ijerph-20-05741-t002:** Trust in Sources for COVID-19 Information, by Race and Ethnicity (n = 1441).

Trust Sources	Hispanic, White (n = 454)	Non-Hispanic, Black (n = 427)	Hispanic, Black (n = 560)	Total (n = 1441)	*p*-Value
	n (%)	n (%)	n (%)	n (%)	
**Doctor or health care provider**					
Not at all	37 (8.2)	32 (7.5)	62 (11.1)	131 (9.1)	**0.037**
A little	189 (41.6)	154 (36.1)	189 (33.8)	532 (36.9)	
A great deal	228 (50.2)	241 (56.4)	309 (55.2)	778 (54.0)	
**Faith leader** **(e.g., pastor/priest)**					
Not at all	76 (16.7)	74 (17.3)	76 (13.6)	226 (15.7)	0.171
A little	202 (44.5)	174 (40.8)	269 (48.0)	645 (44.8)	
A great deal	176 (38.8)	179 (41.9)	215 (38.4)	570 (39.6)	
**Close friends** **/family members**					
Not at all	44 (9.7)	50 (11.7)	50 (8.9)	144 (10.0)	**<0.001**
A little	144 (31.7)	192 (45.0)	216 (38.6)	552 (38.3)	
A great deal	266 (58.6)	185 (43.3)	294 (52.5)	745 (51.7)	
**People you go to work/class with**					
Not at all	54 (11.9)	75 (17.6)	62 (11.1)	191 (13.3)	**<0.001**
A little	180 (39.7)	202 (47.3)	305 (54.5)	687 (47.7)	
A great deal	220 (48.5)	150 (35.1)	193 (34.5)	563 (39.1)	
**News on radio** **/TV/online** **/newspapers**					
Not at all	52 (11.5)	69 (16.2)	50 (8.9)	171 (11.9)	**<0.001**
A little	165 (36.3)	200 (46.8)	286 (51.1)	651 (45.2)	
A great deal	237 (52.2)	158 (37.0)	224 (40.0)	619 (43.0)	
**Contacts on social media**					
Not at all	58 (12.8)	117 (27.4)	66 (11.8)	241 (16.7)	**<0.001**
A little	185 (40.8)	180 (42.2)	314 (56.1)	679 (47.1)	
A great deal	211 (46.5)	130 (30.4)	180 (32.1)	521 (36.2)	
**The U.S.** **Government**					
Not at all	44 (9.7)	61 (14.3)	37 (6.6)	142 (9.9)	**<0.001**
A little	142 (31.3)	210 (49.2)	252 (45.0)	604 (41.9)	
A great deal	268 (59.0)	156 (36.5)	271 (48.4)	695 (48.2)	
**The U.S.** **Coronavirus Task Force**					
Not at all	45 (9.9)	44 (10.3)	36 (6.4)	125 (8.7)	**<0.001**
A little	135 (29.7)	153 (35.8)	147 (26.3)	435 (30.2)	
A great deal	274 (60.4)	230 (53.9)	377 (67.3)	881 (61.1)	

Bold indicates statistical significance.

**Table 3 ijerph-20-05741-t003:** Statistically Significant Covariate Predictors of COVID-19-related Trust.

Trust Questions	Covariate Predictors of Trust vs. the Reference Group	Odds Ratio (95% CI) of Bestowing Higher Level of Trust	*p*-Value for Testing Similarity of the OR to that of the Reference Group	Interpretation
How much do you trust your doctor/health provider	18–30 y vs. 40+ 30–40 y vs. 40+ women vs. men	0.3 (0.2, 0.5) 0.3 (0.2, 0.5) 0.2 (0.2, 0.4)	<0.000 <0.0001 0.0078	Separately, younger and women respondents are 70–80% less likely to trust their doctor/health care
How much do you trust your faith leader	College/higher vs. HS	0.6 (0.5, 0.8)	0.0008	Responders with college degree or higher are 40% less likely to trust their faith leader
How much do you trust your friends and family members	18–30 y vs. 40+ Latinx/non-Latinx, Black vs. Latinx, White women vs. men	1.8 (1.2,2.6) 0.4 (0.3,0.6) 0.4 (0.2,0.7)	0.0038 <0.0001 <0.0001	Younger respondents are 1.8-fold more likely to trust their friends, but women and AA/Blacks are each 60% less likely to trust their friends.
How much do you trust News on the radio/TV/online	Latinx/non-Latinx, Black vs. Latinx, White College/higher vs. HS	0.6 (0.4, 0.9) 0.7 (0.5, 0.9)	0.0188 0.009	Latinx/non-Latinx, Blacks and higher educated respondents are 30–40% less likely to trust news/media and internet news
How much do you trust people you go to work or class with or other people you know	18–30 y vs. 40+ 30–40 y vs. 40+ Latinx/non-Latinx, Black vs. Latinx, White Latinx, other race vs. Latinx, White College/higher vs. HS	2.3 (1.6, 3.4) 2.1 (1.4, 3.0) 0.7 (0.4, 0.9) 0.3 (0.1, 0.9) 0.7 (0.5, 0.9)	<0.0001 0.0002 0.0247 0.0281 0.005	While younger respondents are more than twice as likely to trust people at work/class, Latinx/non-Latinx, Blacks and higher educated respondents are between 70% and 30% less likely to bestow that trust
How much do you trust your contacts in social media	30–40 y vs. 40+ Latinx/non-Latinx, Black vs. Latinx, White women vs. men	1.9 (1.3, 2.8) 0.6 (0.4, 0.8) 0.7 (0.6, 0.9)	0.0006 0.0045 0.0136	While younger respondents are 90% more likely to trust their contacts in social media, Latinx /non-Latinx, Blacks and women are between 30 and 40% less likely to trust them.
How much do you trust the US government	Latinx/non-Latinx, Black vs. Latinx, White women vs. men College/higher vs. HS	0.5 (0.4, 0.8) 0.7 (0.5, 0.8) 0.7 (0.5, 0.98)	0.0007 0.0009 0.0337	Separately, Latinx /non-Latinx, Blacks, women and higher educated respondents are between 50 and 30% less likely to trust the US government
How much do you trust the US Coronavirus Task Force	women vs. men	0.7 (0.6, 0.95)	0.0197	Women are 30% less likely to trust the US coronavirus Task

**Table 4 ijerph-20-05741-t004:** Statistically Significant Covariate Predictors of COVID-19-related Knowledge.

Statements	Covariate/Predictors vs. the Reference Group	Odds Ratio (95% CI) of Knowing the Correct Answer	*p*-Value for Testing Similarity of the OR to that of the Reference Group	Interpretation
Getting a vaccine protects you and means you cannot get the disease if exposed to it in the future	Age: 18–30 y vs. 40+ 30–40 y vs. 40+	2.2 (1.4, 3.2) 2.8 (1.9, 4.1)	<0.0002 <0.0001	Younger (vs. older) respondents are at least 2× more likely to give the correct answer
Getting a vaccine for a disease means you might get sick from the vaccine itself	Latinx/non-Latinx, Black vs. Latinx, White College/higher vs. HS	0.60 (0.4, 0.8) 0.7 (0.5, 0.96)	0.003 0.0272	Latinx/non-Latinx, Black (vs. Latinx, White) and highly educated responders are, respectively, 1.7 and 1.4-fold more likely to answer this question correctly
Hydroxychloroquine is an effective treatment for COVID-19	18–30 y vs. 40+ 30–40 y vs. 40+	2.5 (1.6, 4.0) 2.7 (1.7, 4.1)	<0.0001 <0.0001	Younger (vs. older) respondents are at least more than 2.5-fold more likely to give the correct answer

**Table 5 ijerph-20-05741-t005:** Statistically Significant Covariates Predictors of COVID-19 Testing/Vaccination.

Covariate/Predictor	Testing Odds Ratio (95% CI) *p*-Value	Interpretation	Vaccination Odds Ratio (95% CI) *p*-Value	Interpretation
COVID-19 spreads through coughing and sneezing (correct vs. not correct answer)	1.4 (0.9, 2.1) 0.06	Those who answered this question correctly were (marginally significant) 1.4× more likely to get tested (the sample is generally young and analysis adjusted for age, gender, employment, education, and race)	1.3 (0.99, 1.6) 0.058	Those who answered this question correctly were (marginally significant) 1.3× more likely to get vaccinated (the sample is generally young and analysis adjusted for age, gender employment, education, and race)
Wearing a face mask helps in mitigating the spread of COVID-19 (correct vs. not correct answer)	1.7 (1.2, 2.5) 0.0078	Those who answered this question correctly were 1.7× more likely to get tested	N.S.	N/A
Wearing a face mask may be harmful to your health	0.6 (0.5, 0.9) 0.0192	Those who did not answer this question correctly were 40% less likely to get tested	N.S.	N/A
Employed vs. not employed	2.3 (1.5, 3.4) <0.0001	Those who were employed were 2.3× more likely to get tested	1.8 (2.3, 2.5) 0.0001	Those who were employed were 1.8× more likely to get vaccinated
College/higher vs. High school/lower	1.9 (1.4, 2.7) 0.0002	Those who have college/higher education were 1.9× more likely to get tested	1.5 (1.1, 2.0) 0.00053	Those who have college/higher education were 1.5× more likely to get vaccinated
Latinx/non-Latinx, Black vs. Latinx, White	0.5 (0.33, 0.84) <0.0001	Latinx/non-Latinx, Blacks were 50% less likely to get tested	0.6 (0.4, 0.8) 0.004	Latinx/non-Latinx, Blacks compared to Latinx, Whites were 40% less likely to get vaccinated
Hydroxychloroquine is an effective treatment for COVID-19	0.6 (0.4, 0.9) 0.0104	Those who did not answer this question correctly were 40% less likely to get tested	0.7 (0.5, 0.9) 0.0006	Those who answered this question incorrectly were 30% less likely to get vaccinated
In the U.S. COVID-19 has affected Black Hispanic/ Latino and Native American populations at a higher rate than White populations	0.7 (0.5, 0.9) 0.0188	Those who answered this question incorrectly were 30% less likely to get tested	0.7 (0.6, 0.97) 0.002	Those who answered this question incorrectly were 30% less likely to get vaccinated
Perception of COVID-19 Risk (high vs. low)	N.S.	N/A	2.3 (1.2, 4.4) 0.01	Those who perceived the risk of COVID-19 as high (vs. those who perceived it as low) were 2.3× more likely to get vaccinated
Health Insurance	2.6 (1.7, 4) <0.0001	Those who had health insurance (vs. those who did not) were 2.6× more likely to get tested	2.8 (1.8, 4.3) <0.0001	Those who had health insurance (vs. those who did not) were 2.8× more likely to get vaccinated

## Data Availability

Data associated with this study are not publicly available due to privacy or ethical restrictions delineated in the Institutional Review Board approved protocol.

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
