# Peer review of "Community-Centered Assessment to Inform Pandemic Response in Georgia (US)"

_ijerph, 2023, doi:10.3390/ijerph20095741_

Round 1

Reviewer 1 Report

Thank you for the opportunity to review this work to detail processes, initial findings, and implications associated with the GEORGIA CEAL survey. I appreciated the efforts made by the authors. The work would benefit from some improvements, which I describe below.

The manuscript is well written. But the exclusion & inclusion criteria should be added to the methodology section. the aim of work in the introduction section should be registered as a paragraph and bullets should be removed.

Author Response

Reviewer 1

Thank you for the opportunity to review this work to detail processes, initial findings, and implications associated with the GEORGIA CEAL survey. I appreciated the efforts made by the authors. The work would benefit from some improvements, which I describe below.

The manuscript is well written. But the exclusion & inclusion criteria should be added to the methodology section.

Thank you for this feedback. The exclusion and inclusion criteria are in the Materials and Methods section. (see Page 4)

the aim of work in the introduction section should be registered as a paragraph and bullets should be removed

Thank you for this feedback. This narrative has been reformatted accordingly. (see Page 3)

Reviewer 2 Report

the authors addressed a much discussed topic in the scientific landscape. I offer some suggestions: - the bibliography is pertinent but I suggest that you also consult the studies by Simione et al (2022) How Implicit Attitudes toward Vaccination Affect Vaccine Hesitancy and Behaviour: Developing and Validating the V-IRAP. Int. J. Environ. Res. Public Health, and Simione et al. 2021 where the implicit and explicit attitude of acceptance of the vaccine is measured. trust in medicine and theories about Covid-19 are also considered. the results of these studies could help the authors in the discussion of their results.
- T
he bibliographic sources should be increased given that the topic has been treated by many studies - the section on limitations should be increased: the choice not to use the measure with scales of attitudes, choices or other psychological variables has conditioned the choices of analysis and the scope of the results. Participants have responses to items and not to questionnaires with internal validity
- F
or example, precisely due to the nature of the chosen tool, the authors may not have applied a mediation model which is instead widely used in this topic.
In fact, vaccine hesitancy does not have constant and direct predictors, but many psychological variables act as mediators (see Murphy, el. (2021).  

Author Response

the authors addressed a much discussed topic in the scientific landscape. I offer some suggestions: - the bibliography is pertinent but I suggest that you also consult the studies by Simione et al (2022) How Implicit Attitudes toward Vaccination Affect Vaccine Hesitancy and Behaviour: Developing and Validating the V-IRAP. Int. J. Environ. Res. Public Health, and Simione et al. 2021 where the implicit and explicit attitude of acceptance of the vaccine is measured. trust in medicine and theories about Covid-19 are also considered. the results of these studies could help the authors in the discussion of their results.

Response: Thanks for this feedback. We have strategically updated the references per your guidance and as reflected in the revisions to the manuscript (see Pages 13-14)

- The bibliographic sources should be increased given that the topic has been treated by many studies - the section on limitations should be increased: the choice not to use the measure with scales of attitudes, choices or other psychological variables has conditioned the choices of analysis and the scope of the results. Participants have responses to items and not to questionnaires with internal validity. For example, precisely due to the nature of the chosen tool, the authors may not have applied a mediation model which is instead widely used in this topic. In fact, vaccine hesitancy does not have constant and direct predictors, but many psychological variables act as mediators (see Murphy, el. (2021).  

Response: We appreciate the feedback from the reviewer and completely agree with the reasoning why we did not appeal to mediation analyses for vaccine hesitancy/willingness. However, this constraint is added to the discussion section of limitations. (see Page 14)

Reviewer 3 Report

The authors have carried out a questionnaire study of ideas about, and attitudes to, the recent coronavirus pandemic. Many of the conclusions are, at least in part, predictable, but this study is extremely comprehensive, and may act as a reference point to future studies. The paper is written in accurate and informative language, and it is a pleasure to read. I particularly like the “Interpretation” columns in the tables: this is an extremely useful innovation. I have very few constructive criticisms.

-          Line 3: I can work out what the authors mean when they indicate that they have studied the “South”, but in an International journal, the reader ought not to have to make assumptions based upon a vague understanding of local social geography. For example, in my country, the South is where wealthy people live.

-          Line 52-54: The word “Latinx” is a neologism that appeared in the United States of America within the last two or three decades, and will be obscure to readers in a country such as mine, which has very few individuals of Latino or Latina origin. I fully approve of the use of the gender neutral term “Latinx”, but these lines might include a phrase about the fact that it is being used as a substitute for Latino and Latina.

-          Table 3, top row: The phrase is in the first column, strictly speaking, are not “questions“, but “statements”. The use of the term “statement” will be very familiar to users of questionnaires, and probably preferred by most

-          Table 4, Row 3, Column 5: I think the word “leader” has been inadvertently omitted from the end of the interpretive statement

-          Lines 377-9: There is a font size error

-          Line 433: The word “References” appears at the end of the text, but appears to be redundant

Author Response

The authors have carried out a questionnaire study of ideas about, and attitudes to, the recent coronavirus pandemic. Many of the conclusions are, at least in part, predictable, but this study is extremely comprehensive, and may act as a reference point to future studies. The paper is written in accurate and informative language, and it is a pleasure to read. I particularly like the “Interpretation” columns in the tables: this is an extremely useful innovation. I have very few constructive criticisms.

-          Line 3: I can work out what the authors mean when they indicate that they have studied the “South”, but in an International journal, the reader ought not to have to make assumptions based upon a vague understanding of local social geography. For example, in my country, the South is where wealthy people live.

      Thank you for this comment. We have replaced “the South” with “Georgia (United States)”

-          Line 52-54: The word “Latinx” is a neologism that appeared in the United States of America within the last two or three decades, and will be obscure to readers in a country such as mine, which has very few individuals of Latino or Latina origin. I fully approve of the use of the gender neutral term “Latinx”, but these lines might include a phrase about the fact that it is being used as a substitute for Latino and Latina.

Thank you very much for this comment. Your recommendation has been woven into update narrative in lines 52-52.

-          Table 3, top row: The phrase is in the first column, strictly speaking, are not “questions“, but “statements”. The use of the term “statement” will be very familiar to users of questionnaires, and probably preferred by most

      Thank you very much for this comment. We have replaced the term “questions” to “statements.”

-          Table 4, Row 3, Column 5: I think the word “leader” has been inadvertently omitted from the end of the interpretive statement

 Thank you for this comment. The word leader has been added to complete the statement

-          Lines 377-9: There is a font size error

 Thank you. We have addressed this issue.

-          Line 433: The word “References” appears at the end of the text, but appears to be redundant

      Thank you, the duplicated word has been removed.